# Structure–Activity Relationships of the Antimalarial Agent Artemisinin 10. Synthesis and Antimalarial Activity of Enantiomers of *rac*-5β-Hydroxy-d-Secoartemisinin and Analogs: Implications Regarding the Mechanism of Action

**DOI:** 10.3390/molecules26144163

**Published:** 2021-07-08

**Authors:** Mohamed Jahan, Francisco Leon, Frank R. Fronczek, Khaled M. Elokely, John Rimoldi, Shabana I. Khan, Mitchell A. Avery

**Affiliations:** 1Department of BioMolecular Sciences, Division Medicinal Chemistry, School of Pharmacy, University of Mississippi, University, MS 38677, USA; jahan@olemiss.edu (M.J.); jrimoldi@olemiss.edu (J.R.); 2Department of Drug Discovery and Biomedical Sciences, University of South Carolina, Columbia, SC 29208, USA; jleon@mailbox.sc.edu; 3Department of Chemistry, Louisiana State University, Baton Rouge, LA 70803, USA; ffroncz@lsu.edu; 4Department of Pharmaceutical Chemistry, Tanta University, Tanta 31527, Egypt; kelokely@temple.edu; 5Department of Chemistry, Institute for Computational Molecular Science, Temple University, Philadelphia, PA 19122, USA; 6National Center for Natural Product Research, University of Mississippi, University, MS 38677, USA; skhan@olemiss.edu

**Keywords:** artemisinin, racemate, enantiomers, Mosher ester, diastereomers, *Plasmodium falciparum*, iron

## Abstract

An efficient synthesis of *rac*-6-desmethyl-5β–hydroxy-d-secoartemisinin **2**, a tricyclic analog of *R*-(+)-artemisinin **1**, was accomplished and the racemate was resolved into the (+)-**2b** and (−)-**2a** enantiomers via their Mosher Ester diastereomers. Antimalarial activity resided with only the artemisinin-like enantiomer *R*-(−)-**2a**. Several new compounds **9**–**16**, **19a**, **19b**, **22** and **29** were synthesized from *rac*-**2** but the C-5 secondary hydroxyl group was surprisingly unreactive. For example, the formation of carbamates and Mitsunobu reactions were unsuccessful. In order to assess the unusual reactivity of **2**, a single crystal X-ray crystallographic analysis revealed a close intramolecular hydrogen bond from the C-5 alcohol to the oxepane ether oxygen (O-11). All products were tested in vitro against the W-2 and D-6 strains of *Plasmodium falciparum*. Several of the analogs had moderate activity in comparison to the natural product **1**. Iron (II) bromide-promoted rearrangement of **2** gave, in 50% yield, the ring-contracted tetrahydrofuran **22**, while the 5-ketone **15** provided a monocyclic methyl ketone **29** (50%). Neither **22** nor **29** possessed in vitro antimalarial activity. These results have implications in regard to the antimalarial mechanism of action of artemisinin.

## 1. Introduction

Malaria continues to be one of the most lethal parasitic diseases due to its prevalence in remote, economically challenged regions about the tropical belt. Historically, natural products have played a crucial role in the pharmacotherapy of this disease. The emergence and spread of malarial resistance to conventional antimalarials has been a cause for great concern, but the discovery by Chinese researchers of qinghaosu, the active ingredient of the medicinal herb qinghao (*Artemisia annua*) over four decades ago, was both outstanding and timely. Now referred to as Artemisinin (**1**), this tetracyclic endoperoxylactone (Figure 1) has nanomolar potency against drug-resistant strains of *P. falciparum* such as the W2-Indochina (chloroquine resistant) and D6-Sierra Leone (mefloquine resistant) clones.

The unusual tetracyclic structure of **1** containing a peroxidic 1,2,4-trioxane moiety is known to play an essential role in the antimalarial mechanism of action of artemisinin [1]. Interaction of the endoperoxide moiety with heme, or free Fe (II) leads to the formation of free radical intermediates [2,3,4,5,6,7,8,9,10,11]. The importance of these free radicals is still not fully understood but is widely believed to result in parasitic death [12,13]. The mechanism of action of peroxides towards *Plasmodium falciparum* is a matter of debate with general acceptance that ferrous iron is required to generate carbon free radicals, but a metal-free mechanism has been suggested [14]. The fate of the carbon radicals formed by Fe(II) promoted fragmentation of the artemisinin peroxide is complicated by intermolecular vs. intramolecular processes where clearly not every peroxide-Fe(II) reaction leads to parasite death. Heme-artemisinin adducts have been detected in malaria infected mice and a covalent adduct of manganoporphyrin with a fragmented carbon radical from artemisinin has been reported [15]. The interactions of artemisinin or peroxides with achiral radical promoters (i.e., Fe(II), heme) should be independent of peroxide chirality. A sample of simplified peroxide enantiomers was shown to have equal potency in vitro [16,17]. Interestingly, artemisinin has been shown to form a thioether adduct with cysteine in the presence of Fe(II), ex vivo [3,18,19,20]. More recently, protein targets for the antimalarial effects of artemisinin have been reported [21] and the calcium channel SERCA, *pf*ATP6, has been shown to be a target of artemisinin [22] as has been the translationally controlled tumor protein *pf*TCTP [23,24,25]. Both protein targets have been the subject of protein–drug computational modeling, neither have bound Fe(II), and *pf*TCTP has been shown to form a covalent adduct with artemisinin.

Due to the undesired physiochemical properties of artemisinin, scientists around the globe over the past four decades have established different methodologies to synthesize artemisinin derivatives at different positions (C-3, 6, 7, 9, 10, O-11, O-13 and 16) of the artemisinin skeleton [26,27,28,29,30,31,32,33,34,35,36,37,38,39]. A number of partial analogs substituted at C-4 have been subjected to SAR [40] but the C-5 position has remained synthetically elusive due to the absence of accessible functional groups at or nearby these positions. In regard to total synthetic attempts in our hands, substitution of the side-chain carbon ultimately becoming C5 was unsuccessful except with a cyano moiety [41]. Avery et al., have synthesized numerous modified analogs of artemisinin which include C-13 carbon, N-11 derivatives, C-3 (alkyl, arylalkyl and carboxyalkyl) analogs, C-7 (-OR), C-9 and corresponding 10-deoxo derivatives such as 9β-16-(arylalkyl)-10-deoxoartemisinins. Many of these compounds were more active than artemisinin with improved physiochemical properties, and they present a valuable contribution towards structure–activity relationship (SAR) and quantitative structure–activity relationships (QSAR) [36,38,42,43,44,45,46,47,48,49]. The synthesis of *rac*-6-desmethyl-5β–hydroxy-d-secoartemisinin **2** (a 1,2,4-trioxane) afforded a starting point upon which to conduct [50] heretofore unexplored SAR about the C-5 position. Additionally, as the C-4 position plays a quintessential mechanistic role in supporting a C-4 radical, C-5 derivatives were hypothesized and synthesized in order to test the effect of stabilization of the C-4 radical. Finally, the racemic C-5 alcohol was derivatized as a mixture of two diastereomers to be separated and by bioassay, test whether chirality is important to biological activity.

## 2. Results

### 2.1. Chemistry

Synthesis of the racemic 5-alcohol proved more difficult than reported [50] and required further synthetic studies to produce gram quantities for derivatization and subsequent bioassay. The reported route to **2** was attempted but modified as follows (Scheme 1).

The reported synthesis was modified specifically as follows: ethyl acetoacetate was ketalized under Dean-Stark conditions with ethylene glycol and *p*-toluenesulfonic acid monohydrate (*p*-TsOH·H_2_O) in benzene with azeotropic removal of water to produce 2-carbethoxymethyl-2-methyl-dioxolane **3** [51] which was then reduced to the aldehyde **4** using diisobutylaluminum hydride (DIBAL-H). Temperature control, solvent and mixing rates were crucial to the success of this reduction.

The aldehyde underwent anti-selective aldol condensation with the lithium enolate of cyclohexanone to form the expected alcohol **5**. The labile β-hydroxy group of **5** was protected as the tert-butyldimethylsilyl ether **6** which then underwent direct conversion to the spiro-epoxide **7** via the Corey–Chaykovsky reaction [52]. This labile, unexpectedly volatile epoxide was then ring opened using ethereal hydrogen peroxide and the metal catalyst [53] sodium molybdate dihydrate (Na_2_MoO_4_·2H_2_O)/glycine) affording the unstable hydroperoxide **8**. Finally, simultaneous deprotection, dehydration, desilylation and ring closure were affected readily in a one-pot reaction with *p*-TsOH·H_2_O in undried dichloromethane (DCM) to give racemic **2** as a stable, crystalline solid [50] in 32% yield (Scheme 1).

#### 2.1.1. Ester Derivatives

In our efforts to find a suitable method for synthesis of ester derivatives of **2**, 1.2 molar equivalents of N-(3-dimethylaminopropyl)-N′-ethylcarbodiimide hydrochloride (EDC) [54] were added to a stirring mixture of *rac*-**2**, 1.2 molar equivalents of valeric acid and 19 mol-% of 4-(dimethylaminopyridine) (DMAP) in dry DCM. The progress of the reaction was monitored by TLC analysis and showed slow product formation. After 4 days, additional equivalents of all reagents were added, and the reaction mixture was stirred for another 24 h but unreacted starting material remained. The mixture was allowed to stir for 1 h at 40 °C and worked up. Compound **9** was obtained only in 11% yield as illustrated in Table 1 and unreacted starting material *rac*-**2** was recovered. In continuation of our efforts to find a suitable method for synthesis of an ester derivative from **2**, 2 molar equivalents of benzoyl chloride were added to a suspension of *rac*-**2** and a 2 molar equivalents of both N-methylimidazole (NMI) and N,N,N′,N′-tetramethylethylenediamine (TMEDA) in dry acetonitrile (CH_3_CN) at 0 °C to r.t., for 24 h [55]. It was hoped that the highly basic conditions would result in a competition of the intramolecular H-bond for the diamine. However, reaction progress as monitored by TLC analysis was minimal. When one molar equivalent of all reagents were added and the reaction mixture left to stir for 24 h, **10** was obtained in 42% yield as illustrated in Table 1. When the above condition was modified by adding 4 molar equivalents benzoyl chloride to a suspension of **2** and 4 molar equivalents of both trimethylamine (Et_3_N) and DMAP in dry CH_3_CN at 0 °C to r.t., for 24 h, **10** was afforded in 70% yield. When the same reaction was repeated but using 4 molar equivalents of TMEDA instead of Et_3_N, the yield was improved slightly to afford **10** in 74% yield. Thus, we decided to use the protocol of 4 molar equivalents DMAP/TMEDA, which was used for the rest of the esters in Table 1.

#### 2.1.2. Ketone Derivative

Different methods were applied to oxidize *rac*-**2** to ketone **15**. A mild condition including use a combination of Oxone (potassium peroxymonosulfate) and TEMPO (2,2,6,6-tetramethylpiperidin-1-yl)oxidanyl offered an efficient catalyst system to oxidize secondary alcohol **2** to ketone **15**. Two different solvents (toluene and DCM) were used to find the optimal condition for the oxidation process. Additionally, tetrabutylammonium bromide (Bu_4_NBr) was added as a bromide ion source. The bromide ion was oxidized to hypobromous acid (HOBr) by Oxone and the HOBr promptly oxidizes the nitroxyl radical to the ultimate oxidant, an *N*-oxo ammonium ion [56]. Upon addition of 1 mol% of TEMPO and 2.2 molar equivalents of Oxone to a solution of the *rac*-**2** and 4 mol% of Bu_4_NBr in toluene, the mixture stirred at r.t., for 5 days to afford **15** in 40% yield with some unreacted alcohol **2**. The yield was enhanced to 50% by changing the solvent to DCM and increasing the mol% of both TEMPO and Bu_4_NBr to 10 mol% and 3 molar equivalents of Oxone. However, the reaction time was the same as before i.e., 5 days as illustrated in Table 2. To increase the yield and decrease the reaction time, and keeping in mind to retain the crucial trioxane system, we investigated the use of the well-known Cr(VI) based oxidant, pyridinium chlorochromate (PCC). When *rac*-**2** was dissolved in DCM and added to a suspension of 1.5 molar equivalents of PCC in DCM at r.t., after 48 h, **15** was obtained in 55% yield [57].

#### 2.1.3. Exomethylene Derivative

Using an excess of both base and phosphonium salt was successful in converting **15** to a methylene olefin but was not the desired **16 [58]**. Upon the addition of a solution of **15** to a mixture of 9.5 molar equivalents of both potassium bis(trimethylsilyl)amide (KHMDS) and methyltriphenylphosphonium bromide CH_3_P(C_6_H_5_)_3_Br in dry THF at 0 °C to r.t., for 12 h, an olefin hoped to be **16** was obtained in 24% (Scheme 2).

Tebbe olefination [59] was unsuccessful in converting **15** to *exo*-olefin **16**; instead, decomposed fractions were obtained. In retrospect, it was hoped that using a transition metal based reagent would occur more rapidly than radical decomposition of **15**/**18** but this was not the case.

Attempted Wittig reaction of ketone **15** led to enolization and beta-elimination. The hydroperoxy group could fragment to the alcohol **18x** which added back to the enone (Scheme 3) and the nonperoxy ketone **15x** could then undergo eventual olefination leading to **16x**. Partial structures having a methylene moiety were also seen along with the product **16x**, but none were peroxidic.

#### 2.1.4. Separation of the Two Enantiomers 2a and 2b

Previous studies from the literature of separate enantiomeric peroxides suggested that chirality was unimportant for bioactivity [16,17]. It was felt that this could not be true in all cases because clear bioorganic and biological evidence for the intermediacy of protein receptors exists [21,22,23,24,25]. Separating the two enantiomers of the artemisinin-like *rac*-5-alcohol into (3*R*, 5*R*, 5a*R*, 9a*S*) and (3*S*, 5*S*, 5a*S*, 9a*R*) diastereomers led to the discovery that most if not all of the antimalarial activity was associated with the (3*R*, 5*R*, 5a*R*, 9a*S*) diastereomer. This diastereomer overlays with the X-ray structure of the natural product, R-(+)-artemisinin (the 3*R* trioxane).

##### Mosher Ester Analysis

Enantiopure (*R*)-(−)-α-methoxy-α-(trifluoromethyl)phenylacetyl chloride (*R*)-(−)-MTPA-Cl or Mosher’s acid chloride was used as a chiral derivatizing agent in order to create diastereomeric, alpha-methoxy-alpha-trifluoromethylphenylacetic acid (MTPA) esters **19a** and **19b** upon reaction with *rac*-**2**. These diastereomers were separated by silica gel column chromatography whereupon basic hydrolysis was applied to **19a** and **19b** to obtain the enantiopure **2a** and **2b**, respectively. In addition, optical activity of **2a** and **2b** were compared. Addition of 2 molar equivalents of (*R*)-(−)-MTPA-Cl to a stirring suspension of *rac*-**2** and 3 molar equivalents of both DMAP and TMEDA in dry CH_3_CN [54] at 0 °C to r.t., for 72 h was examined by TLC (silica gel with 20% EtOAc/hexane solvent and *p*-anisaldehyde stain), and showed reaction completion. Two separable spots of high *R*_f_ (0.63) and low *R*_f_ (0.58) of **19b** and **19a** was observed in 19% and 22%, respectively. 

Hydrolysis of the two Mosher’s esters **19a** and **9b** occurred smoothly without destruction of the essential trioxane system. They were treated with a solution of 1N sodium hydroxide (NaOH) in methanol (MeOH) [59] at ambient temperature for 36 h, furnishing **2a** and **2b** in 27 and 24% yield, respectively (Scheme 4). Furthermore, the specific rotations, [α]D^20^ were calculated for **2a** and **2b** and found to be ‒100.5° and +100.5°, respectively.

#### 2.1.5. Rearrangement Chemistry

Within the malaria parasite, heme Fe (II) (reduced hemin) or other sources of ferrous iron can induce chemical decomposition of artemisinin and related tricyclic trioxanes (e.g., *rac*-**2**, **1****5**) to generate an oxy radical that subsequently rearranges into distinctive carbon-centered radical species. (e.g., **20** and **21**; Scheme 5 and Scheme 6). Using FeBr_2_ as a mimic of heme iron(II), we performed ferrous-mediated degradation of racemic **2** and **15**. When Fe(II) associated with oxygen 1, an oxy radical **20** was formed and rearrangement occurs to provide a primary carbon-centered radical **21** via a C_3_-C_4_ scission process. This intermediate radical species then undergoes ring-contraction to the tetrahydrofuran product **22** (Scheme 5). As mentioned earlier, this reactive intermediate (e.g., **21**) could be responsible for alkylation of biomacromolecules such as heme, specific proteins, and other targets that cause the death of malaria parasites.

Upon treatment *rac*-**2** at ambient temperature in THF with 2 equivalents of FeBr_2_ for 2 h [45], **22** was isolated by flash chromatograph in 47% yield as the only tractable product from rearrangement of *rac*-**2**. Assignment of a structure to **22** was based on spectral evidence; it showed in its IR spectra strong absorption bands at 3394 and 1740 cm^−1^ corresponding to hydroxyl and acetyl groups, respectively. In the NMR, the presence of a methyl of the acetyl group was clearly indicated as a singlet at δ 2.11. In addition, HMBC and HSQC experiments confirmed the connectivity presented for compound **22**. For example, the C-8 protons at δ 4.62 (dd, *J* = 1.5, 12.3 Hz, 1H, CH_2a_-8), and 4.26 (d, *J* = 12.3 Hz, 1H, CH_2**b**_-8), and the methyl at δ 2.11 (s, 3H, CH_3-_11), showed strong HMBC correlations to C-10, a carbonyl assigned to δ 171.8. Furthermore, the quaternary carbon at δ 81.8 showed strong HMBC correlations with the protons at CH-3, CH_2_-8 and CH_2_-2, confirming the THF ring in the molecule **22**. The nuclear Overhauser effect spectroscopy (NOESY) experiment showed a correlation between the protons at CH-3 and CH-3a indicating a cis configuration between these two protons.

When ketone **15** was treated with 2 molar equivalents of FeBr_2_ in THF, for 2 h, **29** was isolated by flash chromatograph in 50% yield as the only tractable product from the rearrangement of **15**. Plausible mechanisms for generation of **29** are illustrated in paths (a) and (b). Association of Fe (II) with oxygen-1 leads to an oxy radical that scissions to the primary carbon-centered radical **23.** The stabilized enol radical **23** abstracts a hydrogen atom from THF and generates **24** and **25**. Involvement of a trace of water (path a) leads to protonated (tetrahydrofuran-2-yl)oxonium ion **26** that then quenches the Fe(II)oxo intermediate **27** and generates 2-hydroxytetrahydrofuran **28** and the (2-acetyl-1-hydroxycyclohexyl)methyl acetate **29**, respectively. In case of truly anhydrous THF, **24** could lose a proton (path b) to protonate the Fe(II)oxo intermediate **27** thus generating **29** and 2,3-dihydrofuran **30** (Scheme 6). 

Product **29** showed strong hydroxyl and carbonyl absorption in the IR at 3452, 1740 and 1705 cm^−1^ and was confirmed in the proton NMR spectra which indicated the presence of an acetyl group at *δ* 2.08 and methyl ketone at *δ* 2.23. In addition, HMBC and HSQC experiments confirmed the connectivity shown for compound **29**. For instance, the C-7 protons at δ 4.24 (d, *J* = 12.0 Hz, 1H, CH_2a_-7), and 4.18 (d, *J* = 12.0 Hz, 1H, CH_2b_-7), and the methyl at δ 2.08 (s, 3H, CH_3-_10), showed a strong HMBC correlations to C-9, a carbonyl assigned to *δ* 171.8. The C-2 carbon at *δ* 56.4 showed strong correlations with the protons at CH_2_-7 and the methyl signal at *δ* 2.23 (s, 3H, CH_3-_12) confirming the presence of the methyl ketone. Unfortunately, the 5-exomethylene derivative **16** could not be prepared for this study but may have given a similar result to **15**. 

### 2.2. Antimalarial Activity 

Artemisinin **1** and the analogs *rac*-**2**, **2a**, **2b**, **9**–**16**, diastereomers **19a**, **19b**, **22** and **29** were tested in vitro in parasitized whole blood against drug-resistant strains of *P*. *falciparum* parasite clones D-6 and W-2 at The National Center for Natural Products Research (NCNPR) at The University of Mississippi using the parasite lactate dehydrogenase (pLDH) assay developed by Makler (Table 3) [60,61,62]. This assay is based on the ability of the pLDH enzyme of *P. falciparum* to reduce APAD to APADH. This reaction is carried out at a slow rate by human red blood cell LDH.

The formation of APADH was monitored colorimetrically by the addition of nitroblue tetrazolium which was reduced to a blue formazan product. Two *P. falciparum* malaria parasite clones, designated as Indochina (W-2) and Sierra Leone (D-6), were used in susceptibility testing. The W-2 clone is chloroquine-resistant and mefloquine-sensitive, while the D-6 clone is chloroquine-sensitive but mefloquine resistant. The relative potency values for these analogs were derived from the IC_50_ value for artemisinin (**1**) divided by their IC**_50_** values (Table 3) and then adjusted for molecular weight differences by multiplication of the ratio of the molecular weight of the analog divided by the molecular weight of artemisinin. This approach to reporting activity was based in part on the fact that the analogs were tested on different occasions in which the IC_50_ for the control, artemisinin, had varied anywhere from 5 to 7 ng/mL based on parasitemia levels and clone.

### 2.3. X-ray Crystallography

A single crystal X-ray diffraction analysis was conducted to verify the structure of *rac-***2**. The crystal used was not in a racemic space group and was a pure enantiomer. The racemate likely crystallized as a *conglomerate*, that is the (+) and (−) enantiomers formed separate crystals, from which **2b** was chosen randomly. The X-ray confirms that the crystal structure of **2b** is an orthorhombic (P2_1_ 2 _l_ 2 _l_) with cell dimensions of a = 5.7823 Å, b = 7.6135 Å, and c =23.795 Å. The total volume of the unit cell is 1047.5 Å. It also confirmed that a hydrogen bond was present between H4O and O1 with a total length of 1.9 Å. In addition, a crystal structure was also generated for better visualization utilizing commercially available molecular software (Sciencomics, MAPS 3.4) and was drawn as **2b**. The *cif coordinates file (generated from the X-ray analysis) was imported to molecular dynamics software and gave the atomic representation (Figure 2a). The atomic structure is represented by the following color code: hydrogen white balls, carbon is grey balls and oxygen is blue balls. The hydrogen bond is also evident as represented by the dashed line (Figure 2b). 

## 3. Discussion

Antimalarial activities of the new analogues synthesized are presented in Table 3. The relative activities range from 0.62% (D-6) and 0.43% (W-2) in case of *rac*-**2** to 33.30% (D-6) and 32.72% (W-2) in case of the ketone **15**. We were not surprised to find that enantiopure (‒)-**2a** was responsible for the antimalarial activity with relative activities of 0.43% (D-6) and 0.44% (W-2), values almost the same as *rac*-**2**; while enantiopure (+)-**2b** was devoid of antimalarial activity. Of the ester derivatives, 4-fluoro ester **11** was the most active with relative activities of 20.81% (D-6) and 12.41% (W-2) while the enantiomer **19b** was the least active with relative activities of 0.50% (D-6) and 0.34% (W-2). The Fe (II)-induced rearrangement products **22** and **29** had no antimalarial activity due to loss of the natural product-like 1,2,4-trioxane ring.

The ketone **15** and exomethylene **16** were designed taking into consideration likely rearrangement behavior relative to alcohols or esters derived from **2**. We were thus pleased to find that the ketone **15** was 55–77 times more potent than the alcohol *rac*-**2** or **2a** (i.e., **2**). Structures **15** and **2** have very similar overall shape due to their highly constrained tricyclic systems. The only major difference between the alcohol **2** and ketone **15** is the intramolecular H-bond in **2**. While there may be an effect on the initial O-O homolytic cleavage of **2** due to the IMH-bond, relative to **15**, a more important difference is likely related to the intermediate alkylating species. C-radical **21** should be less stable than C-radical **23** (Scheme 5 and Scheme 6) and furthermore, we know that radical **21** undergoes intramolecular ring closure quenching the carbon radical while carbon radical **23** is forced to react in an intermolecular sense, avoiding self-immolation. In the active site of a protein target, **23** produced from **15** should be longer-lived than **21** from **20**. It might be argued that **23** is more active than **21** because **23** would have a better chance to react in an intermolecular sense than **21** and, thus, would be more likely to alkylate protein. While this is only one example of this theory, synthesis of **16** could be a laudable goal.

Analysis of **2a** by DFT calculations (Jaguar, Schrödinger 21-1 release; see Figure 3 and Figure 4) revealed the importance of the intramolecular H-bond in stabilizing the structural energy. Two structures, local minimum **2a** no-IMHB, and local minimum **2a**-IMHB 2-up were apparent with a ΔE of 5.1 kcal/mol. However, a global minimum revealed **2a**-IMHB 2-down, identical to the X-ray structure **2b** (Figure 2 and Figure 3), was 10.4 kcal/mol more stable than **2a** without an IMHB. The **2a**-IMHB 2-down boat overlaps with the unstable trioxane (+)-artemisinin1down (Figure 4). The large difference between H-bonded and the non–H-bonded conformers corresponds to an exclusive intramolecular H-bond (IMHB) in **2a**-IMHB-2-down. The significance of this H-bond in dictating the reactivity of *rac*-**2** was unforeseen during its synthesis. Once its abnormal properties became apparent with *rac*-**2** in hand, we hoped reaction solvent changes and/or additives could have effected the extent of IMHB, but we were disconcerted by the unreactive nature of *rac*-**2** as became apparent vide supra. 

Continuing the analysis of conformational effects, we noticed that two ring-flipped variants existed for artemisinin, both trioxane rings adopting a boat conformation. One of these, the lower E variant, we referred to as O1-UP because O1 appears higher than O2. As shown in Figure 4, this is illustrated as (+)-artemisinin1UP and the local minimum (+)-artemisinin1DOWN was also obtained and it had a ΔE of 12.56 kcal/mol relative to Artemisinin1UP, the later representing the global minimum and corresponding to the conformation found in the X-ray structure of the natural product. One can see that the effect of the IMHB in **2a is** to ring-flip the trioxane to 2-down with a ΔE of 10.4 kcal/mol. No other artemisinin derivatives have the “1DOWN” conformation as seen with the benzoate **10.** The ketone **15**, having increased rigidity, transmits conformational effects to the peroxide to effect a ring flip to **15** 2-down. How this preference for a natural trioxane conformation in **2a**-IMHB 2-down and ketone **15** 2-down compare to unnatural **10** 2-up in regards to rearrangement/radical chemistry and thence, antimalarial activity awaits further study.

Overall, these comparisons show that **2a** exists in an exclusive IMH-bonded conformation by X-ray analysis and DFT calculations. Thus, the poor or absent reactivity of alcohol **2** towards many acylation partners must be related to a reticent H-bond, and ultimately inhibited a more detailed SAR/QSAR. With more **2a** analogs in hand, application of QM parameterization to QSAR could reveal the effects of peroxide geometry on antimalarial activity. Several reports [63,64,65,66,67,68] described the application of different quantum mechanical approaches to studying the geometrical parameters and chemical reactivity descriptors of artemisinin. As the valence-separate basis set 6-31G** of DFT/B3LYP method showed the best results as suggested by Santos et al. [69], we implemented this technique in our calculations.

## 4. Materials and Methods

### 4.1. Chemistry

#### 4.1.1. General Information 

Optical rotations were recorded using Rudolph Research Analytical Autopol V Polarimeter. Melting points were measured on an OptiMelt^®^ V.1061 (Stanford Research systems) instrument and were uncorrected. ^1^H and ^13^C NMR spectra were obtained on Bruker NMR spectrometers model DRX 600, DRX 500 and DRX 400 NMR spectrometers with standard pulse sequences, operating at 600, 500 and 400 MHz in ^1^H and 150, 125 and 100 MHz in ^13^C, respectively. The chemical shifts values were reported in parts per million units (ppm) from trimethylsilane (TMS) using known solvent CDCl_3,_ C_6_D_6_ chemical shifts. Coupling constants were recorded in Hertz (Hz), standard pulses were used for COSY, HSQC, HMBC, and NOESY experiments. Infrared (IR) spectra were recorded on a PerkinElmer Spectrum 100FT-IR Spectrometer. High resolution mass spectra (HRMS) were measured with a Waters Q-TOF Micromass spectrometer using the electrospray ionization (ESI) source in negative or positive mode. Flash chromatography was performed using silica gel (Whatman 60Å, 230–400 mesh). Analytical thin layer chromatography (TLC) was performed on EMD Chemical INC 25 TLC aluminum sheets, silica gel 60 F_254_ or GP Analtech TLC plates. All reaction solvents were purchased as HPLC grade and, where appropriate, distilled from CaH_2_ and then stored over 3 or 4 Å molecular sieves. Most commercial reagents were used without further purification unless otherwise noted in the procedure. All reagents and dry solvents were purchased from Sigma-Aldrich, Fluka, or Thermo Fisher Scientific. All round bottom flasks were dried properly in a vacuum oven prior to reactions. Solvents and reagent transfers were accomplished via dried syringes or cannulas. All reactions were performed under argon atmosphere, unless otherwise specified.

General information this: The NMR spectra and crystalographic data are available in the Appendix A.

#### 4.1.2. Preparation and Characterization of Synthesized Compounds

##### Ethyl 2-(2-methyl-1,3-dioxolan-2-yl) acetate (**3**)

A solution of ethyl acetoacetate (150.0 g, 1.15 mol), ethylene glycol (214 g, 3.44 mol), and *p*-TsOH·H_2_O (9.70g, 51 mmol) in 1100 mL of benzene in a 2 L round bottom flask was refluxed for 5 h with continuous azeotropic separation of H_2_O via a Dean-Stark trap. The mixture was cooled to ambient temperature, concentrated to 500 mL in vacuo and then washed sequentially with sat. aq. NaHCO_3_ (2 × 100 mL) and brine (100 mL), filtered over anhydrous Na_2_SO_4_ and evaporated in vacuo. The crude material was distilled under vacuum (20–25 mbar) at 109–110 °C to afford **3** (167.3 g, 83%) as a colorless oil; ^1^H-NMR (400 MHz, CDCl_3_) δ: 4.09 (q, *J* = 7.1 Hz, 2H), 3.91 (s, 4H), 2.59 (s, 2H), 1.43 (s, 3H), 1.20 (t, *J* = 7.1 Hz, 3H) ppm; ^13^C-NMR (100 MHz, CDCl_3_) δ: 169.3, 107.5, 64.7, 60.4, 44.1, 24.4, 14.1 ppm; IR (neat): 2984, 2890, 1732, 1447,1369, 1318, 1240,1183, 1096, 1040, 949 cm^−1^; ESI-HRMS: calcd for C_8_H_15_O_4_ [M + H]^+^: 175.0900, found: 175.0973.

##### 2-(2-Methyl-1,3-dioxolan-2-yl)acetaldehyde (**4**)

To a 1000 mL 2-necked round-bottomed flask equipped with argon line and septa dry DCM (250 mL) and the ester **3** (55.09 g, 0.32 mol) were added. The mixture was vigorously stirred at −78 °C for 15 min, whereupon DIBAL-H (57.92.18 g, 0.41 mol, 1.0 M solution in toluene) was added dropwise via cannula and the reaction mixture was then stirred at −78 °C for 5 h. The reaction was checked by TLC for completion, and then quenched with MeOH (40 mL). The mixture was allowed to warm to ambient temperature and stirred for 40 min. A saturated aqueous solution of Rochelle salt (200 mL) was added and the reaction mixture was stirred overnight. The aqueous layer was separated and extracted with DCM (3 × 120 mL). The combined organic layers were washed with brine (2 × 150 mL) and dried over anhydrous MgSO_4_ and then the crude product was filtered through a pad of silica gel atop a pad of celite and the clear solution was concentrated by rotary evaporation. The crude product was distilled under vacuum (41mbar) at 98–100 °C to afford **4** (31.6 g, 76%) as a pale yellow oil; ^1^H-NMR (400 MHz, CDCl_3_) δ: 9.73 (t, *J* = 2.9 Hz, 1H), 3.99 (m, 4H), 2.70 (d, *J* = 2.9 Hz, 2H), 1.41 (s, 3H) ppm; ^13^C-NMR (100 M Hz, CDCl_3_) δ: 200.2, 107.6, 64.8, 52.2, 24.9 ppm; IR (neat): 2986, 2889, 1721, 1380, 1293, 1148, 1115, 1050, 948, 863 cm^‒1^; ESI-HRMS calcd for C_6_H_16_NaO_3_ [M + Na]^+^: 153.0528, found: 153.1689.

##### 2-(1-Hydroxy-2-(2-methyl-1,3-dioxolan-2-yl)ethyl)cyclohexan-1-one (**5**) 

To a 300 mL round-bottomed flask equipped with argon line and septum *n*-BuLi (3.63 g, 56 mmol, 2.5 M solution in hexanes) was added to a −78 °C solution of *i-*Pr_2_NH (5.73 g, 56 mmol) in dry THF (150 mL). The mixture was warmed to 0 °C and stirred for 30 min. and then the bath temperature was lowered to −78 °C. A solution of cyclohexanone (5.05 g, 51 mmol) in dry THF (40 mL) was added dropwise via cannula. The mixture was stirred at −78 °C for 2 h at which time the aldehyde **4** (6.71 g, 51 mmol) was added via syringe and the mixture stirred for 1 h at −78 °C. The reaction was quenched with glacial acetic acid (6.19 g, 103 mmol) and the bath was allowed to warm to ambient temperature naturally and the reaction mixture was stirred overnight. Saturated aqueous NH_4_Cl (100 mL) was added, followed by Et_2_O (120 mL). Phases were separated and the aqueous layer back-extracted with Et_2_O (3 × 100 mL). The combined organic layers were washed with water and brine (2 × 100 mL) and dried over anhydrous Na_2_SO_4_. After filtration, the solvent was removed by rotary evaporation. The reaction product was then chromatographed over silica gel with 30% EtOAc /hexane to afford **5** as a colorless oil (7.8 g, 66%); ^1^H-NMR (400 MHz, CDCl_3_) δ: 4.17 (m, 1H), 3.97 (br s, 4H), 3.54 (d, *J* = 2.8 Hz, 1H), 2.49 (dt, *J* = 11.3, 5.4 Hz, 1H), 2.41–2.25 (m, 2H), 2.13 (m, 1H), 2.03 (m, 1H), 1.86 (m, 2H), 1.81–1.62 (m, 3H), 1.51 (m, 1H), 1.39 (s, 3H) ppm; ^13^C-NMR (100 MHz, CDCl_3_) δ: 213.1, 110.2, 67.5, 64.6, 64.2, 55.7, 42.5, 41.9, 29.3, 27.7, 24.6, 24.0 ppm; IR (neat): 3519, 2935, 2867, 1702, 1449, 1405, 1378, 1313, 1250, 1201, 1128, 1111, 1050, 948, 835, 818 cm^‒1^; ESI-HRMS calcd for C_12_H_20_NaO_4_ [M + Na]^+^: 251.1259, found: 251.1264.

##### 2-(1-((Tert-butyldimethylsilyl)oxy)-2-(2-methyl-1,3-dioxolan-2-yl)ethyl)cyclohexan-1-one (**6**)

To a 100 mL round-bottomed flask equipped with argon line and septum, a solution of alcohol **5** (2.6 g, 11.3 mmol) in dry DMF (30 mL) was added and stirred at 0 °C for 5 min. Then, 2,6-Lutidine (5.9 g, 55 mmol) was added via syringe and the reaction mixture was stirred for 20 min. TBSCl (2.51 g, 16.7 mmol) was added to the above reaction mixture followed by 22 mol% of DMAP and stirred at 0 °C for 3 h and then for 12 h at ambient temperature. Saturated aqueous NH_4_Cl (20 mL) was added followed by Et_2_O (80 mL). Phases were separated and the aqueous layer back-extracted with Et_2_O (3 × 50 mL). The combined organic layers were washed with water (100 mL) and brine (2 × 100 mL) and dried over anhydrous Na_2_SO_4_. Solvent was removed by rotary evaporation. The crude product was purified by silica gel flash chromatography (30% EtOAc/hexane) to give **6** as a colorless oil (2.82, 72%); ^1^H-NMR (400 MHz, CDCl_3_) δ: 4.37 (dt, *J* = 3.2, 7.2 Hz, 1H), 4.02–3.87 (m, 4H), 2.46 (m, 1H), 2.33 (m, 1H), 2.25–2.15 (m, 2H), 2.01–1.84 (m, 3H), 1.70–1.52 (m, 4H), 1.41 (s, 3H), 0.88 (s, 9H), 0.07 (d, *J* = 17.0 Hz, 6H) ppm; ^13^C-NMR (100 MHz, CDCl3) δ: 211.3, 109.3, 67.6, 64.3, 64.0, 56.7, 42.3, 41.8, 26.9, 26.8, 25.9, 24.8, 24.7, 18.0, −4.5 ppm; IR (neat): 2931, 2856, 1708, 1472, 1378, 1251, 1165, 1113, 1046, 1030, 923, 830, 773 cm^‒1^; ESI-HRMS calcd for C_18_H_34_NaO_4_Si [M + Na]^+^: 365.2124, found: 365.2111.

##### Tert-butyldimethyl(2-(2-methyl-1,3-dioxolan-2-yl)-1-(1-oxaspiro[2.5]octan-4-yl)ethoxy)silane (**7**)

To a 100 mL round-bottomed flask equipped with argon line and septum, a suspension of NaH (0.83 g, 34 mmol, in 60% (w/w) mineral oil) in dry DMSO (30 mL) was added and stirred at ambient temperature for 5 h, then the mixture was added via cannula to a suspension of Me_3_SI (2.40 g, 11 mmol) in dry DMSO (15 mL) and dry THF (15 mL) stirred at 0 °C. The mixture was stirred at 0 °C for 40 min. A solution of ketone **6** (2.02 g, 5.9 mmol) in dry THF (12 mL) was added to the above mixture and stirred for 20 min at 0 °C. The mixture was then allowed to warm to ambient temperature and stirred for 15 h before being partitioned between Et_2_O (100 mL) and brine (60 mL). Phases were separated and the aqueous layer back-extracted with Et_2_O (3 × 50 mL). The combined organic layers were washed with water (100 mL) and brine (100 mL) and dried over anhydrous Na_2_SO_4_. Solvent was removed by rotary evaporation. The crude product was purified by silica gel flash chromatography (30% EtOAc/hexane) and the relevant fractions were combined. Normal rotary evaporative solvent removal at 20 mm Hg led to lower yields with some product in the condensate. More careful evaporation afforded the epoxide **7** as a light, volatile yellow oil (1.14 g, 54%). Again, significant loss of epoxide occurred to evaporation and could provide a means of improvement in yield. ^1^H-NMR (400 MHz, CDCl_3_) δ: 3.91 (s br, 4H), 3.03 (d, *J* = 4.3 Hz, 1H), 2.41 (d, *J* = 4.3 Hz, 1H), 2.06 (d, *J* = 14.6 Hz, 1H), 1.94 (m, 1H), 1.92–1.80 (m, 2H), 1.79–1.60 (m, 2H), 1.54–1.38 (m, 2H), 1.34 (s, 3H), 1.30–1.17 (m, 4H), 0.89 (s, 9H), 0.08 (s, 6H) ppm; ^13^C-NMR (100 MHz, CDCl_3_) δ: 109.8, 68.6, 64.2, 64.0, 61.0, 50.4, 45.5, 41.5, 36.2, 25.9, 25.7, 24.8, 24.3, 23.6, 18.1, ‒3.9, ‒4.3 ppm; IR (neat): 2930, 2856, 1472, 1375, 1252, 1170, 1100, 1037, 944, 771 cm^−1^; ESI-HRMS calcd for C_19_H_36_NaO_4_Si [M + Na]^+^: 379.2281, found: 379.2276.

##### (2-(1-((tert-butyldimethylsilyl)oxy)-2-(2-methyl-1,3-dioxolan-2-yl)ethyl)-1hydroperoxycyclohexyl)methanol (**8**)

To a 100 mL round-bottomed flask equipped with argon line and septum, a solution of epoxide **7** (1.80 g, 5.1 mmol) in ethereal H_2_O_2_ (Et_2_O washed with saturated NaCl 90% H_2_O_2_, 50 mL) was added and stirred at ambient temperature for 5 min before the catalyst (Na_2_MoO_4_·2H_2_O and glycine, 0.21 g, 0.5 mmol) was added. The mixture was stirred for 24 h, whereupon water (60 mL) was added to the reaction mixture which was then extracted with EtOAc (3 × 80 mL). The combined organic layers were washed with water and brine (100 mL) and dried over anhydrous Na_2_SO_4_. Solvent was removed by rotary evaporation to afford the unstable hydroperoxide **8** as a light-yellow oil (1.20 g, 61%).

##### 3-Methyloctahydro-1h-3,9a-epidioxybenzo[c]oxepin-5-ol, (rac-2)

To a 150 mL round-bottomed flask **8** (1.15 g, 2.9 mmol), DCM (80 mL) and *p*-TsOH·H_2_O (0.28 g, 1.47 mmol) were added. The mixture was stirred at ambient temperature for 24 h. Saturated aqueous NaHCO_3_ (100 mL) was added, then the reaction mixture was extracted with Et_2_O (3 × 125 mL). The combined organic layers were washed with water (100 mL) and brine (100 mL) and dried over anhydrous Na_2_SO_4_. Solvent was removed by rotary evaporation and the crude product was chromatographed on silica gel (20% EtOAc/hexane) to afford the *rac*-**2** as a crystalline solid (0.20 g, 32%); m.p. 109–111 °C; ^1^H-NMR (400 MHz, C_6_D_6_) δ: 4.02 (d, *J* = 11.3 Hz, 1H, CH_2a_-10), 3.86 (d, *J* = 11.3 Hz, 1H, CH_2b_-10), 3.75 (m, 1H, CH-5), 3.64 (d, *J* = 11.9 Hz, 1H, OH), 2.70 (dd, *J* = 3.2, 15.0 Hz, 1H, CH_2a_-4), 2.35 (dd, *J* = 3.1, 15.0 Hz, 1H, CH_2b_-4), 1.75 (m, 1H, CH_2a_-7), 1.67–1.62 (m, 3H, CH-5a, CH_2b_-7, CH_2a_-9), 1.52 (d br, *J* = 12.8 Hz, 1H, CH_2a_-6), 1.32 (m, 1H, CH_2a_-8), 1.31 (s, 3H, CH_3-_12), 1.21 (td, *J* = 4.5, 13.7 Hz, 1H, CH_2b_-9), 0.98 (m, 1H, CH_2b_-6), 0.81 (tt, *J* = 3.6, 13.4 Hz, 1H, CH_2a_-8) ppm; ^13^C-NMR (100 MHz, C_6_D_6_) δ: 105.1 (C-3), 82.8 (C-9a), 68.3 (C-5), 63.5 (C-10), 51.9 (C-5a), 44.9 (C-4), 35.2 (C-9), 27.1 (C-7), 26.2 (C-12), 25.5 (C-6), 23.6 (C-8) ppm; IR (neat): 3413, 2928, 2856, 1463, 1361, 1252, 1069, 1006, 938, 850, 771 cm^−1^; ESI-HRMS calcd for C_11_H_18_NaO_4_ [M + Na]^+^: 237.1103, found: 237.0937; C_11_H_19_O_4_ [M + H]^+^: 215.1283, found: 215.1059.

##### 3-Methyloctahydro-1H-3,9a-epidioxybenzo[c]oxepin-5-yl pentanoate (**9**)

To a 10 mL round-bottom flask equipped with argon line and septum a solution of *rac*-**2** (16 mg, 0.075 mmol) in dry DCM (2mL), valeric acid (0.09 mmol, 9.83 µL), EDC (0.09 mmol, 17.17 mg) and DMAP (1.82 mg, 0.015 mmol) was added. After stirring at ambient temperature for 4 days, additional valeric acid (9.83 µL), EDC (8 mg) and DMAP (4 mg) were added and stirred continuously for another 24 h. The reaction mixture was then stirred at 40 °C for 1 h. The mixture was filtered, and the filtrate was collected and diluted by DCM (15 mL). Subsequently, the filtrate was washed by 0.1 N HCl (10 mL), saturated aqueous NaHCO_3_ (10 mL) and brine 10 (mL), dried over anhydrous Na_2_SO_4_, concentrated by rotary evaporation and the crude product was chromatographed on silica gel (20% EtOAc/hexane) to afford **9** (2.5 mg 11%); ^1^H-NMR (400 MHz, CDCl_3_) *δ*: 5.26 (s br, CH-5), 4.24 (d, *J* = 10.4 Hz, 1H, CH_2a_-10), 4.05 (dd, *J* = 0.9, 10.4 Hz, 1H, CH_2b_-10), 2.70 (dd, *J* = 3.2, 15.8Hz, 1H, CH_2a_-4), 2.38–2.30 (m, 3H, CH_2_-2′, CH_2b_-4), 2.06 (m, 1H, CH-5a), 1.87 (d br, *J* = 12.5 Hz, 1H, CH_2a_-9), 1.78–1.60 (m, 5H, CH_2a_-6, CH_2a_-7, CH_2a_-8, CH_2_-3′), 1.49 (m, 1H, CH_2b_-6), 1.30–1.20 (m, 6H, CH_2b_-9, CH_3_-12, CH_2b_-7, CH_2b_-8, CH_2_-4’), 0.93 (m, 3H, CH_3_-5′) ppm; ^13^C-NMR (100 MHz, CDCl_3_) δ: 173.6 (C-1′), 103.0 (C-3), 82.5 (C-9a), 69.4 (C-5), 63.4 (C-10), 49.9 (C-5a), 43.1 (C-4), 35.8 (C-9), 34.5 (C-2′), 27.1 (C-7), 26.2 (C-3′), 26.1 (C-12), 25.3 (C-6), 23.6 (C-8), 22.3 (C-4´), 13.7 (C-5′) ppm; ESI-HRMS calcd for C_16_H_26_CsO_5_ [M + Cs]^+^: 431.0835, found: 431.0945.

##### 3-Methyloctahydro-1H-3,9a-epidioxybenzo[c]oxepin-5-yl benzoate (**10**)

To a 5 mL round-bottom flask equipped with argon line and septum *rac*-**2** (20 mg, 0.093 mmol), CH_3_CN (1 mL), DMAP (45 mg, 0.37 mmol) and TMEDA (56 µL, 0.37 mmol) were added. The mixture was stirred at 0 °C for 10 min. Then benzoyl chloride (43 µL, 0.37 mmol) was added and the reaction mixture stirred for 1 h. The mixture was allowed to warm to ambient temperature and stirred for 24 h. Water (2 mL) was added to the stirred mixture, which was extracted with EtOAc (15 mL). The organic phase was washed with brine (10 mL), dried over anhydrous Na_2_SO_4_, and concentrated by rotary evaporation. The crude product was purified by flash chromatography over silica gel (30% EtOAc/hexane) to give **10** as a white solid (22 mg, 74%); m.p. 100–102 °C; ^1^H-NMR (400 MHz, CDCl_3_) *δ*: 8.07 (d br, *J* = 7.4 Hz, 2H, CH-2′ and CH-6´), 7.57 (t br, 7.8 Hz, 1H, C-4´), 7.46 (t, *J* = 7.8 Hz, 2H, CH-3′ and CH-5′), 5.55 (ddd, *J* = 3.1, 6.4, 6.3 Hz, C-5), 4.37(d, *J* = 10.9 Hz, 1H, CH_2a_-10), 4.14 (dd, *J* = 1.7, 10.8 Hz, 1H, CH_2b_-10), 2.81 (dd, *J* = 3.5, 15.8Hz, 1H, CH_2a_-4), 2.45 (dd, *J* = 2.9, 15.8 Hz, 1H, CH_2b_-4), 2.15 (m, 1H, CH-5a), 1.91 (d br, *J* = 12.2 Hz, 1H, CH_2a_-9), 1.78–1.66 (m, 3H, CH_2a_-6, CH_2a_-7, CH_2a_-8), 1.52 (td, *J* = 3.4, 12.9 Hz, 1H, CH_2b_-6), 1.37 (m, 1H, CH_2b_-9), 1.35 (s, 3H, CH_3_-12), 1.21 (m, 2H, CH_2b_-7, CH_2b_-8) ppm; ^13^C-NMR (100 MHz, CDCl_3_) δ: 166.2 (C=O), 132.9 (C-4´), 130.6 (C-1′), 129.7 (C-2′ and C-6´), 128.5 (C-3′ and C-5′), 103.2 (C-3), 82.6 (C-9a), 70.0 (C-5), 63.6 (C-10), 50.3 (C-5a), 43.3 (C-4), 35.8 (C-9), 26.4 (C-6), 26.1 (C-12), 25.2 (C-7), 23.6 (C-8) ppm; IR (neat): 2934, 2864, 1710, 1601, 1449, 1374, 1314, 1274, 1220, 1205, 1183, 1160, 1136, 1111, 1070, 1055, 1024, 966, 91, 887, 838, 821, 783 cm^‒1^; ESI-HRMS calcd for C_18_H_22_CsO_5_ [M + Cs]^+^: 451.0522, found: 451.0526.

##### 3-Methyloctahydro-1H-3,9a-epidioxybenzo[c]oxepin-5-yl 4-fluorobenzoate (**11**)

To a 15 mL round-bottom flask equipped with argon line and septum *rac*-**2** (20 mg, 0.093 mmol), CH_3_CN (5 mL), DMAP (45 mg, 0.37 mmol) and TMEDA (56 µL, 0.37 mmol) were added. The mixture was stirred at 0 °C for 10 min. Then, 4-fluorobenzoyl chloride (44 µL, 0.37 mmol) was added and the reaction mixture stirred for 1 h. The mixture was allowed to warm to ambient temperature and stirred for 48 h. Water (5 mL) was added to the stirred mixture, which was extracted with EtOAc (20 mL). The organic phase was washed with brine (20 mL), dried over anhydrous Na_2_SO_4_, and concentrated by rotary evaporation. The crude product was purified by flash chromatography over silica gel (20% EtOAc/hexane) to give **11** as a light yellow solid (22 mg, 70%); m.p. 88–90 °C; ^1^H-NMR (400 MHz, CDCl_3_) δ: 8.10 (m, 2H, CH-2′ and CH-6´), 7.13 (t br, *J* = 8.6 Hz, 2H, CH-3′ and CH-5′), 5.55 (m, C-5), 4.36 (d, *J* = 11.0 Hz, 1H, CH_2a_-10), 4.15 (dd, *J* = 1.8, 11.0 Hz, 1H, CH_2b_-10), 2.82 (dd, *J* = 3.6, 15.8 Hz, 1H, CH_2a_-4), 2.44 (dd, *J* = 2.9, 15.8 Hz, 1H, CH_2b_-4), 2.14 (m, 1H, CH-5a), 1.91 (d br, *J* = 13.1 Hz, 1H, CH_2a_-9), 1.79–1.69 (m, 3H, CH_2a_-6, CH_2a_-7, CH_2a_-8), 1.46 (m, 1H, CH_2b_-6), 1.38 (m, 1H, CH_2b_-9), 1.36 (s, 3H, CH_3_-12), 1.20 (m, 2H, CH_2b_-7, CH_2b_-8) ppm; ^13^C-NMR (100 MHz, CDCl_3_) δ: 165.8 (C-4´; *J*_FC_ = 252.4 Hz), 165.2 (C=O), 132.2 (C-2′ and C-6´; *J*_FC_ = 9.2 Hz), 126.9 (C-1′; *J*_FC_ = 2.8 Hz), 115.5 (C-3′ and C-5′; *J*_FC_ = 21.8Hz), 103.2 (C-3), 82.6 (C-9a), 70.2 (C-5), 63.6 (C-10), 50.3 (C-5a), 43.3 (C-4), 35.8 (C-9), 26.4 (C-6), 26.1 (C-12), 25.3 (C-7), 23.6 (C-8) ppm; IR (neat): 2931, 2862, 1704, 1603, 1507, 1453, 1410, 1374, 1279, 1223, 1151, 1110, 1016, 962, 856, 801, 767 cm^‒1^; ESI-HRMS calcd for C_18_H_21_FNaO_5_ [M + Na]^+^: 359.1271, found: 359.2250.

##### 3-Methyloctahydro-1H-3,9a-epidioxybenzo[c]oxepin-5-yl 4-methoxybenzoate (**12**)

To a 15 mL round-bottom flask equipped with argon line and septum *rac*-**2** (20 mg, 0.093 mmol), CH_3_CN (5 mL), DMAP (45 mg, 0.37 mmol) and TMEDA (56 µL, 0.37 mmol) were added. The mixture was stirred at 0 °C for 10 min. Then, 4-methoxybenzoyl chloride (50 µL, 0.37 mmol) was added and the reaction mixture stirred for 1 h. The mixture was allowed to warm and stir at ambient temperature for 72 h. Water (5 mL) was added to the stirred mixture, which was extracted with EtOAc (20 mL). The organic phase was washed with brine (20 mL), dried over anhydrous Na_2_SO_4_, and concentrated by rotary evaporation. The crude product was purified by flash chromatography over silica gel (20% EtOAc/hexane) to give **12** as a white solid (22 mg, 68%); m.p. 155–157 °C; ^1^H-NMR (400 MHz, CDCl_3_) δ: 8.04 (d br, *J* = 8.8 Hz, 2H, CH-2′ and CH-6´), 6.94 (d br, *J* = 8.8 Hz, 2H, CH-3′ and CH-5′), 5.53 (ddd, *J* = 3.2, 6.4, 6.4 Hz, C-5), 4.37 (d, *J* = 10.9 Hz, 1H, CH_2a_-10), 4.14 (dd, *J* = 1.8, 10.9 Hz, 1H, CH_2b_-10), 3.87(s, 3H, OCH_3_), 2.82 (dd, *J* = 3.6, 15.8 Hz, 1H, CH_2a_-4), 2.43 (dd, *J* = 2.8, 15.8 Hz, 1H, CH_2b_-4), 2.13 (m, 1H, CH-5a), 1.90 (dd br, *J* = 1.9, 11.8 Hz, 1H, CH_2a_-9), 1.77–1.69 (m, 3H, CH_2a_-6, CH_2a_-7, CH_2a_-8), 1.49 (ddd, *J* = 3.5, 12.8, 16.4 Hz, 1H, CH_2b_-6), 1.36 (m, 1H, CH_2b_-9), 1.35 (s, 3H, CH_3_-12), 1.21 (m, 2H, CH_2b_-7, CH_2b_-8) ppm; ^13^C-NMR (100 MHz, CDCl_3_) δ: 165.9 (C=O), 163.4 (C-4´), 131.7 (C-2′ and C-6´), 123.0 (C-1′), 113.7 (C-3′ and C-5′), 103.2 (C-3), 82.6 (C-9a), 69.6 (C-5), 63.6 (C-10), 50.3 (C-5a), 43.4 (C-4), 35.9 (C-9), 26.4 (C-6), 26.1 (C-12), 25.3 (C-7), 23.6 (C-8) ppm; IR (neat): 2923, 2859, 1710, 1604, 1578, 1512, 1451, 1350, 1248, 1276, 1160, 1117, 1101, 1045, 1018, 967, 853, 820, 771 cm^‒1^; ESI-HRMS calcd for C_19_H_24_NaO_6_ [M + Na]^+^: 371.1471, found: 371.1747; C_19_H_24_CsO_6_ [M + Cs]^+^: 481.0627, found: 481.0659.

##### 3-Methyloctahydro-1H-3,9a-epidioxybenzo[c]oxepin-5-yl cinnamate (**13**)

To a 15 mL round-bottom flask equipped with argon line and septum *rac*-**2** (50 mg, 0.23 mmol), CH_3_CN (7 mL), DMAP (114 mg, 0.93 mmol) and TMEDA (139 µL, 0.93 mmol) were added. The mixture was stirred at 0 °C for 10 min. Then, cinnamoyl chloride (155 mg, 0.93 mmol) was added and the reaction mixture stirred for 1 h. The mixture was warmed and allowed to stir at ambient temperature for 72 h. Water (10 mL) was added to the stirred mixture, which was extracted with EtOAc (20 mL). The organic phase was washed with brine (20 mL), dried over anhydrous Na_2_SO_4_, and concentrated by rotary evaporation. The crude product was purified by flash chromatography over silica gel (20% EtOAc/hexane) to give **13** as yellow oil which solidified upon storage in the refrigerator (29 mg, 36%); ^1^H-NMR (400 MHz, CDCl_3_) δ: 7.71 (d, *J* = 16.0 Hz, 1H, CH-3′), 7.55 (s br, 2H, CH-2″ and CH-6″), 7.40 (s br, 3H, CH-3″, CH-5″, CH-4″), 6.49 (d, *J* = 16.0 Hz, 1H, CH-2′), 5.45 (m, 1H, C-5), 4.32 (d, *J* = 10.8 Hz, 1H, CH_2a_-10), 4.11 (d, *J* = 10.8 Hz, 1H, CH_2b_-10), 2.78 (dd, *J* = 3.7, 15.9 Hz, 1H, CH_2a_-4), 2.43 (dd, *J* = 2.8, 15.9 Hz, 1H, CH_2b_-4), 2.10 (m, 1H, CH-5a), 1.90 (d br, *J* = 14.0 Hz, 1H, CH_2a_-9), 1.74–1.65 (m, 3H, CH_2a_-6, CH_2a_-7, CH_2a_-8), 1.54 (m, 1H, CH_2b_-6), 1.47 (m, 1H, CH_2b_-9), 1.36 (s, 3H, CH_3_-12), 1.22 (m, 2H, CH_2b_-7, CH_2b_-8) ppm; ^13^C-NMR (100 MHz, CDCl_3_) δ: 166.8 (C=O), 144.9 (C-3′), 134.4 (C-1″), 130.3 (C-4″), 128.8 (C-3″ and C-5″), 128.1 (C-2″ and C-6″), 118.5 (C-2′), 103.1 (C-3), 82.6 (C-9a), 69.6 (C-5), 63.6 (C-10), 50.1 (C-5a), 43.3 (C-4), 35.8 (C-9), 26.3 (C-6), 26.1 (C-12), 25.3 (C-7), 23.6 (C-8) ppm; IR (neat): 2931, 2863, 1702, 1636, 1449, 1374,1310, 1275, 1202, 1180, 1156, 1130, 1072, 1054, 1024, 979,965, 916, 887, 836, 820, 767, 711 cm^‒1^; ESI-HRMS calcd for C_20_H_24_CsO_5_ [M + Cs]^+^: 477.0678, found: 477.0681.

##### 3-Methyloctahydro-1H-3,9a-epidioxybenzo[c]oxepin-5-yl 4-chlorobenzoate (**14**)

To a 15 mL round-bottom flask equipped with argon line and septum *rac*-**2** (30 mg, 0.14 mmol), CH_3_CN (5 mL), DMAP (68 mg, 0.56 mmol) and TMEDA (83 µL, 0.56 mmol) were added. The mixture was stirred at 0 °C for 10 min. Then, 4-chlorobenzoyl chloride (72 µL, 0.56 mmol) was added and the reaction mixture stirred for 1 h. The mixture was allowed to warm and stir at ambient temperature for 24 h. Water (5 mL) was added to the stirred mixture, which was extracted with EtOAc (20 mL). The organic phase was washed with brine (20 mL), dried over anhydrous Na_2_SO_4_, and concentrated by rotary evaporation. The crude product was purified by flash chromatography over silica gel (20% EtOAc/hexane) to give **14** as white solid (24 mg, 49%); m.p. 110–112 °C; ^1^H-NMR (400 MHz, CDCl_3_) δ: 8.02 (d br, *J* = 8.6 Hz, 2H, CH-2′ and CH-6´), 7.44 (d, *J* = 8.6 Hz, 2H, CH-3′ and CH-5′), 5.55 (ddd, *J* = 2.4, 6.0, 6.1 Hz, C-5), 4.35(d, *J* = 11.0 Hz, 1H, CH_2a_-10), 4.15 (dd, *J* = 1.8, 10.9 Hz, 1H, CH_2b_-10), 2.83 (dd, *J* = 3.5, 15.8 Hz, 1H, CH_2a_-4), 2.45 (dd, *J* = 2.8, 15.8 Hz, 1H, CH_2b_-4), 2.16 (m, 1H, CH-5a), 1.92 (dd br, *J* = 1.0, 13.1 Hz, 1H, CH_2a_-9), 1.78–1.65 (m, 3H, CH_2a_-6, CH_2a_-7, CH_2a_-8), 1.46 (ddd, *J* = 3.6, 13.2, 16.4 Hz, 1H, CH_2b_-6), 1.37 (m, 1H, CH_2b_-9), 1.36 (s, 3H, CH_3_-12), 1.22 (m, 2H, CH_2b_-7, CH_2b_-8) ppm; ^13^C-NMR (100 MHz, CDCl_3_) δ: 165.3 (C=O), 139.4 (C-4´), 131.1 (C-2′ and C-6´), 129.0 (C-1′), 128.8 (C-3′ and C-5′), 103.2 (C-3), 82.6 (C-9a), 70.4 (C-5), 63.6 (C-10), 50.2 (C-5a), 43.3 (C-4), 35.8 (C-9), 26.4 (C-6), 26.1 (C-12), 25.3 (C-7), 23.6 (C-8) ppm; IR (neat): 2929, 2860, 1701, 1591, 1487, 1450, 1400, 1368, 1271, 1222, 1161, 1136, 1111, 1088, 1014, 965, 854, 819, 750 cm^‒1^; ESI-HRMS calcd for C_18_H_21_ClCsO_5_ [M + Cs]^+^: 485.0132, found: 485.0169.

##### 3-Methylhexahydro-1H-3,9a-epidioxybenzo[c]oxepin-5(5aH)-one (**15**)

To a 25 mL round-bottom flask equipped with argon line and septum PCC (80 mg, 0.37 mmol) and dry DCM (10 mL) were added. The mixture was stirred at ambient temperature for 5 min before a solution of *rac*-**2** (53 mg, 0.25 mmol) in dry DCM (5 mL) was added via cannula to the above mixture and stirred for 48 h. The mixture was diluted with Et_2_O (10 mL), decanted and concentrated by rotary evaporation. The crude material was purified by flash chromatography over silica gel (20% EtOAc/hexane) to give **15** as a yellow oil which solidified upon storage in the refrigerator (29 mg, 55%); ^1^H-NMR (400 MHz, CDCl_3_) *δ*: 4.14 (d, *J* = 11.4 Hz, 1H, CH_2a_-10), 4.11 (d, *J* = 11.4 Hz, 1H, CH_2b_-10), 3.56 (d, *J* = 15.6 Hz, 1H, CH_2a_-4), 2.93 (d, *J* = 15.6 Hz, 1H, CH_2b_-4), 2.36 (m, CH-5a), 2.32 (m, 1H, CH_2a_-6), 1.92–1.87 (d br, *J* = 14.1 Hz, 1H, CH_2a_-9), 1.80–1.70 (m, 2H, CH_2a_-7, CH_2a_-8), 1.44 (ddd, *J* = 4.4, 13.5, 18.0 Hz, 1H, CH_2b_-9), 1.37 (s, 3H, CH_3_-12), 1.30–1.12 (m, 3H, CH_2b_-6, CH_2b_-7, CH_2b_-8), ppm; ^13^C-NMR (100 MHz, CDCl_3_) *δ*: 204.9 (C-5), 103.0 (C-3), 83.4 (C-9a), 63.6 (C-10), 57.6 (C-5a), 54.9 (C-4), 34.3 (C-9), 25.4 (C-12), 25.0 (C-6), 24.6 (C-7), 23.4 (C-8) ppm; IR (neat): 2935, 2865, 1715, 1450, 1409, 1376, 1335, 1320, 1268, 1247, 1213, 1190, 1167, 1118, 1097, 1055, 909, 881, 863, 827, 778 cm^‒1^; ESI-HRMS calcd for C_11_H_17_O_4_ [M + H]^+^: 213.1127, found: 213.1140.

##### (3*R*,5*R*,5a*R*,9a*S*)-3-methyloctahydro-1H-3,9a-epidioxybenzo[c]oxepin-5-yl(R)-3,3,3-trifluoro-2-methoxy-2-phenylpropanoate (**19a**) and (3*S*,5*S*,5a*S*,9a*R*)-3-methyloctahydro-1H-3,9a-epidioxybenzo[c]oxepin-5-yl(R)-3,3,3-trifluoro-2-methoxy-2-phenylpropanoate (**19b**)

To a 50 mL round-bottom flask equipped with argon line and septum *rac*-**2** (195 mg, 0.91 mmol), CH_3_CN (25 mL), DMAP (334 mg, 2.73 mmol) and TMEDA (0.41 mL, 2.73 mmol) were added. The mixture was stirred at 0 °C for 10 min. Then, (*R*)-(−)-MTPA-Cl (0.34 mL, 1.82 mmol) was added and the reaction mixture stirred for 1 h. The mixture was allowed to warm and stirred at ambient temperature for 72 h. Water (20 mL) was added to the stirred mixture, which was extracted with EtOAc (2 × 25 mL). The organic phases were washed with brine (30 mL), dried over anhydrous Na_2_SO_4_, and concentrated by rotary evaporation. The crude product was purified by flash chromatography over silica gel (15% EtOAc/hexane) to give two diasteromers. The higher *R_f_* material **19b** (76 mg, 19%) was viscous oil; ^1^H-NMR (600 MHz, CDCl_3_) δ: 7.71 (m, 2H, CH-2″ and CH-6″), 7.42 (m, 3H, CH-3″, CH-5″, CH-4″), 5.40 (dt, *J* = 3.3, 9.6 Hz, 1H, C-5), 4.19 (d, *J* = 11.0 Hz, 1H, CH_2a_-10), 4.05 (dd, *J* = 1.7, 11.1 Hz, 1H, CH_2b_-10), 3.65 (s, 3H, OCH_3_), 2.79 (dd, *J* = 3.0, 16.2 Hz, 1H, CH_2a_-4), 2.50 (dd, *J* = 2.8, 16.2 Hz, 1H, CH_2b_-4), 2.10 (m, 1H, CH-5a), 1.88 (d br, *J* = 11.5 Hz, 1H, CH_2a_-9), 1.69–1.63 (m, 2H, CH_2a_-7, CH_2a_-8), 1.54 (m, 1H, CH_2a_-6), 1.34 (s, 3H, CH_3_-12), 1.32 (m, 1H, CH_2b_-9),1.19–1.12 (m, 3H, CH_2b_-6, CH_2b_-7, CH_2b_-8) ppm; ^13^C-NMR (150 MHz, CDCl_3_) δ: 166.1 (C-1′), 132.2 (C-1″), 129.5 (C-4″), 128.3 (C-3″ and C-5″), 127.2 (C-2″ and C-6″), 125.5 (C-3′; *J* _CF_ = 238 Hz), 102.6 (C-3), 83.8 (C-2′), 82.5 (C-9a), 73.2 (C-5), 63.4 (C-10), 55.6 (OCH_3_), 50.2 (C-5a), 42.7 (C-4), 35.7 (C-9), 26.0 (C-6), 26.0 (C-12), 25.4 (C-7), 23.5 (C-8) ppm; ESI-HRMS calcd for C_21_H_25_CsF_3_O_5_ [M + Cs]^+^: 563.0658, found: 563.0663. The lower *Rf* material **19a** (82 mg, 22%) was a viscous oil; ^1^H-NMR (400 MHz, CDCl_3_) δ: 7.56 (m, 2H, CH-2″ and CH-6″), 7.43 (m, 3H, CH-3″, CH-5″, CH-4″), 5.40 (dt, *J* = 3.1, 9.4 Hz, 1H, C-5), 4.06 (d, *J* = 11.0 Hz, 1H, CH_2a_-10), 3.97 (dd, *J* = 1.7, 11.0 Hz, 1H, CH_2b_-10), 3.50 (s, 3H, OCH_3_), 2.76 (dd, *J* = 3.5, 16.1 Hz, 1H, CH_2a_-4), 2.45 (dd, *J* = 2.8, 16.1 Hz, 1H, CH_2b_-4), 2.08 (m, 1H, CH-5a), 1.84 (d br, *J* = 13.6 Hz, 1H, CH_2a_-9), 1.68–1.56 (m, 3H, CH_2a_-6, CH_2a_-7, CH_2a_-8), 1.35–1.26 (m, 2H, CH_2b_-6, CH_2b_-9), 1.24 (s, 3H, CH_3_-12), 1.19–1.10 (m, 2H, CH_2b_-7, CH_2b_-8) ppm; ^13^C-NMR (100 MHz, CDCl_3_) δ: 166.4 (C-1′), 131.7 (C-1″), 129.6 (C-4″), 128.4 (C-3″ and C-5″), 127.8 (C-2″ and C-6″), 127.5 (C-3′; *J* _CF_ = 228 Hz), 102.6 (C-3), 84.7 (C-2′), 82.5 (C-9a), 73.2 (C-5), 63.4 (C-10), 55.1 (OCH_3_), 50.2 (C-5a), 42.7 (C-4), 35.7 (C-9), 26.0 (C-6), 26.0 (C-12), 25.4 (C-7), 23.5 (C-8) ppm; IR (neat): 2940, 2852, 1746, 1460, 1376, 1271, 1161, 1127, 1023, 837, 718 cm^‒1^; ESI-HRMS calcd for C_21_H_25_CsF_3_O_5_ [M + Cs]^+^: 563.0658, found: 563.0691.

##### (3*R*,5*R*,5a*R*,9a*S*)-3-methyloctahydro-1H-3,9a-epidioxybenzo[c]oxepin-5-ol (**2a**)

To a 25 mL round-bottom flask containing **19a** (20 mg, 0.05 mmol), a warm solution of 1N NaOH in MeOH (10 mL) was added. The mixture was allowed to stir at ambient temperature for 36 h. The mixture was then neutralized with a few drops of dilute HCI. After removal of the MeOH by rotary evaporation, the residue was dissolved in CHCl_3_ (10 mL) and washed with H_2_O (10 mL). The aqueous layer was extracted with CHCI_3_ (2 x 10 mL). The combined organic layers were filtered over anhydrous Na_2_SO_4_ and concentrated by rotary evaporation. The crude product was purified by flash chromatography over silica gel (20% EtOAc/hexane) to afford **2a** as a crystalline solid (2.6 mg, 27%); m.p. 108–110 °C; ^1^H-NMR (400 MHz, CDCl_3_) δ: 4.19 (d, *J* = 11.3 Hz, 1H, CH_2a_-10), 4.11 (dd, *J* = 1.4, 11.3 Hz, 1H, CH_2b_-10), 3.84 (m, 1H, CH-5), 3.64 (d, *J* = 11.7 Hz, 1H, OH), 2.74 (dd, *J* = 3.3, 15.2 Hz, 1H, CH_2a_-4), 2.44 (dd, *J* = 3.1, 15.2 Hz, 1H, CH_2b_-4), 1.90–1.58 (m, 6H, CH_2a_-7, CH_2b_-8, CH-5a, CH_2b_-7, CH_2a_-9, CH_2a_-6), 1.40 (s, 3H, CH_3_-12), 1.33–1.20 (m, 3H, CH_2a_-8, CH_2b_-9, CH_2b_-6) ppm; ^13^C-NMR (100 MHz, CDCl_3_) δ: 105.2 (C-3), 83.8 (C-9a), 68.6 (C-5), 63.6 (C-10), 51.7 (C-5a), 44.8 (C-4), 35.1 (C-9), 27.0 (C-7), 26.3 (C-12), 25.4 (C-6), 23.8 (C-8) ppm; [α]_D_
^20^ = ‒100.5° (*c* = 0.016, CHCl_3_); IR (neat): 3426, 2928, 2856, 1462, 1449, 1376, 1361, 1252, 1161, 1048, 1030, 830, 771 cm^‒1^; ESI-MS calcd for C_11_H_17_O_4_ [M − H]^−^: 213.1127, found: 213.0793.

##### (3*S*,5*S*,5a*S*,9a*R*)-3-methyloctahydro-1H-3,9a-epidioxybenzo[c]oxepin-5-ol (**2b**)

To a 25 mL round-bottom flask containing **19b** (24 mg, 0.06 mmol), a warm solution of 1N NaOH in MeOH (10 mL) was added. The mixture was allowed to stir at ambient temperature for 36 h. The mixture was then neutralized with few drops of dilute HCl. After removal of the MeOH by rotary evaporation, the residue was dissolved in CHCl_3_ (10 mL) and washed with H_2_O (10 mL). The aqueous layer was extracted with CHCl_3_ (2 x 10 mL). The combined organic layers were filtered over anhydrous Na_2_SO_4_ and concentrated by rotary evaporation. The crude product was purified by flash chromatography over silica gel (20% EtOAc/hexane) to afford **2b** as a crystalline solid (2.9 mg, 24%); m.p. 107–109 °C; ^1^H-NMR (400 MHz, CDCl_3_) δ: 4.19 (d, *J* = 11.3 Hz, 1H, CH_2a_-10), 4.11 (dd, *J* = 1.4, 11.3 Hz, 1H, CH_2b_-10), 3.84 (m, 1H, CH-5), 3.64 (d, *J* = 11.7 Hz, 1H, OH), 2.74 (dd, *J* = 3.3, 15.2 Hz, 1H, CH_2a_-4), 2.44 (dd, *J* = 3.1, 15.2 Hz, 1H, CH_2b_-4), 1.90–1.58 (m, 6H, CH_2a_-7, CH_2b_-8, CH-5a, CH_2b_-7, CH_2a_-9, CH_2a_-6), 1.40 (s, 3H, CH_3_-12), 1.33–1.20 (m, 3H, CH_2a_-8, CH_2b_-9, CH_2b_-6) ppm; ^13^C-NMR (100 MHz, CDCl_3_) δ: 105.2 (C-3), 83.8 (C-9a), 68.6 (C-5), 63.6 (C-10), 51.7 (C-5a), 44.8 (C-4), 35.1 (C-9), 27.0 (C-7), 26.3 (C-12), 25.4 (C-6), 23.8 (C-8) ppm; [α]_D_
^20^ = +100.5° (*c* = 0.020, CHCl_3_); IR (neat): 3417, 2931, 2862, 1449, 1387, 1357, 1240, 1221, 1160, 1111, 1092, 1030, 984, 903, 882, 853, 832, 816 cm^‒1^; ESI-MS calcd for C_11_H_17_O_4_ [M − H]^−^: 213.1127, found: 213.0741.

##### (3-Hydroxyhexahydrobenzofuran-7a(2H)-yl)methyl acetate (**22**)

To a 5 mL round-bottom flask *rac*-**2** (19 mg, 0.089 mmol), THF (1.2 mL), and FeBr_2_ (0.038 g, 0.177 mmol) were added. The mixture was stirred at ambient temperature under argon atmosphere for 2 h and then directly chromatographed over silica gel (60% EtOAc/hexane). The product **22** was isolated in 47% yield (9 mg); m.p. 123–125 °C; ^1^H-NMR (600 MHz, CDCl_3_) δ: 4.62 (dd, *J* = 1.5, 12.3 Hz, 1H, CH_2a_-8), 4.42 (td, *J* = 1.5, 5.5 Hz, 1H, CH-3), 4.26 (d, *J* = 12.3 Hz, 1H, CH_2b_-8), 4.23 (dd, *J* = 5.2, 10.5 Hz, 1H, CH_2a_-2), 3.85 (dd, *J* = 1.6, 10.5 Hz, 1H, CH_2b_-2), 2.30 (dt, *J* = 3.3 6.7 Hz, 1H, CH_2a_-7), 2.11 (s, 3H, CH_3-_11), 1.91 (m, 1H, CH_2a_-5), 1.83 (m, 1H, CH_2a_-4), 1.76 (m, 1H, CH_2a_-6), 1.54 (m, 1H, CH-3a), 1.46 (m, 1H, CH_2b_-4), 1.35–1.26 (m, 3H, CH_2b_-7, CH_2b_-5, CH_2b_-6,) ppm; ^13^C-NMR (150 MHz, CDCl_3_) δ: 171.8 (C-10), 81.8 (C-7a), 75.2 (C-2), 72.2 (C-3), 61.8 (C-8), 53.4 (C-3a), 33.6 (C-7), 25.6 (C-5), 22.6 (C-6), 21.7 (C-4), 21.0 (C-11) ppm;. IR (neat): 3394, 2944, 2881, 1740, 1481, 1451, 1387, 1368, 1353, 1311, 1242, 1205, 1148, 1124, 1081, 1036, 1018, 1008, 1018, 973, 926, 870, 845, 755 cm^‒1^; ESI-HRMS calcd for C_11_H_18_NaO_4_ [M + Na]^+^: 237.1103, found: 237.1145.

##### (2-Acetyl-1-hydroxycyclohexyl)methyl acetate (**29**)

To a 5 mL round-bottom flask **15** (20 mg, 0.09 mmol), THF (1.2 mL), and FeBr_2_ (0.041 g, 0.177 mmol) were added. The mixture was stirred at ambient temperature under argon atmosphere for 2 h and then directly chromatographed over silica gel (60% EtOAc/hexane). The product **29** was isolated in 50% yield (10 mg); ^1^H-NMR (400 MHz, CDCl_3_) δ: 4.24 (d, *J* = 12.0 Hz, 1H, CH_2a_-7), 4.18, (d, *J* = 12.0 Hz, 1H, CH_2b_-7), 2.68 (dd, *J=* 4.4, 9.8 Hz, 1H, CH-2), 2.23 (s, 3H, CH_3-_12), 2.08 (s, 3H, CH_3-_10), 2.01–1.92 (m, 2H, CH_2a_-6, CH_2a_-3), 1.68–1.59 (m, 3H, CH_2b_-6, CH_2a_-4, CH_2a_-5), 1.43–1.34 (m, 3H, CH_2b_-3, CH_2b_-4, CH_2b_-5) ppm; ^13^C-NMR (100 MHz, CDCl_3_) δ: 211.6 (C-11), 171.8 (C-9), 72.8 (C-1), 67.3 (C-7), 56.4 (C-2), 34.5 (C-6), 30.9 (C-12), 25.2 (C-3), 23.8 (C-4), 22.1 (C-5), 20.8 (C-10) ppm; IR (neat): 3452, 2924, 2854, 1740, 1705, 1366, 1259, 1236, 1186, 1090, 1014, 924, 800, 737 cm^‒1^; ESI-HRMS calcd for C_11_H_18_NaO_4_ [M + Na]^+^: 237.1103, found: 237.1132.

### 4.2. Single Crystal X-ray Diffraction Experiment

#### Compound *rac**-***2**

Racemic **2** crystallized as a *conglomerate*, from which **2b** was randomly selected. Crystal data for C_11_H_18_O_4_ (M = 214.25 g/mol): clear colorless prism (0.38 × 0.30 × 0.16 mm), orthorhombic, space group *P*2_1_2_1_2_1_, *a* = 5.7823 (5) Å, *b* = 7.6135 (10) Å, *c* = 23.795 Å, *v* = 1047.5 Å^3^ and *z* = 4 Å. *d*_calcd_ = 1.359 mg × m^−3^, 4872 independent reflections measured out to *θ*_max_ = 36.6° with a Nonius KappaCCD diffractometer using Mo Kα radiation (*λ =* 0.71073 Å) with a graphite monochromator in the incident beam. The data were collected at room temperature by using the *ω* scan technique. Multi-scan absorption correction was applied using Denzo and Scalepack [70]. The structure was solved by direct methods as implemented by the SHELXTL97 system of programs [70]. Full-matrix least-squares refinement was performed on 141 parameters using the 4872 reflections. The C-H distances were fixed at 0.98–1.00 Å and placed in idealized positions.

### 4.3. Antimalarial Activity Assay

Synthesized compounds were tested in vitro against two *P. falciparum* malaria parasite clones, designated as Indochina (W-2) and Sierra Leone (D-6). The W-2 clone is chloroquine resistant whereas the D-6 clone is a chloroquine-sensitive strain.

#### 4.3.1. Reagents and Materials

The two *P. falciparum* clones, W-2 and D-6 were obtained from Walter Reed Army Institute. Human blood and human serum were obtained from Interstate Blood Bank. Roswell Park Memorial Institute medium (RPMI 1640 medium), acetic acid and 96-well microplate were purchased from Thermo Fisher Scientific. APAD, NBT, PES, artemisinin, chloroquine, DMSO, amikacin, doxorubicin, phosphate-buffered saline (PBS) and neutral red were purchased from Sigma-Aldrich. Vero cells (African green monkey kidney fibro-blast) purchased from American Type Culture Collection (ATCC).

#### 4.3.2. In Vitro Antimalarial Assay

A suspension of red blood cells infected with W-2 or D-6 strain of *P. falciparum* that contains 2% parasitemia and 2% hematocrit in RPMI 1640 medium supplemented with 10% human serum and 60 µg/mL amikacin was dispensed into the wells of a 96-well flat-bottomed microtiter plate containing 10 µL of serially diluted test samples. The plates were incubated at 37 °C in an environment of 90% N_2_, 5% O_2_, and 5% CO_2_ for 72 h. Next, 100 µL aliquots of the Malstat reagent were added to each well of a new 96-well microtiter plate. The cultures were resuspended from the assay plate by mixing each well up and down several times. A total of 20 µL from each well of the resuspended culture was removed and added to the plate containing the Malstat reagent and the plate was incubated at r.t., for 30 min. Further, to each well, 20 µL of a NBT/PES (1:1) solution (2 and 0.1 mg/mL, respectively) were added. The plate was incubated in the dark for 1 h. The reaction was terminated by the addition of 100 µL of a 5% acetic acid solution. The 139 plate was then read at 650 nm. Artemisinin and chloroquine were included in each assay as antimalarial drug controls. The IC_50_ values were computed from the dose–response curves using XLfit 4.2.

#### 4.3.3. In Vitro Cytotoxicity Assay

The assay was performed in 96-well tissue culture-treated plates as described earlier. Vero cells were seeded to the wells of 96-well plate at a density of 25,000 cells/well and incubated for 24 h. Tested compounds at different concentrations were added and plates were again incubated for 48 h. The number of viable cells was determined by Neutral Red assay [71,72]. IC_50_ values were obtained from dose–response curves. Doxorubicin was used as a positive control for cytotoxicity.

##### Neutral Red Assay

Briefly, after incubating with the tested compounds, the cells were washed with PBS and incubated for 90 min with the medium containing Neutral Red (166 μg/mL). The cells were washed to eliminate extracellular dye. A solution of acidified isopropanol (0.33% HCl) was then added to lyse the cells. As a result, the absorbed dye was released from the viable cells. The absorbance was read at 540 nm. IC_50_ (the concentration of the test compounds that caused a growth inhibition of 50% after 48 h of exposure of the cells) was calculated from the dose curves created by plotting percent growth vs. the test concentration on a logarithmic scale using Microsoft Excel. All assays were performed in triplicate.

## 5. Conclusions

The 5β-hydroxy-secoartemisinin *rac*-**2** proved to be resistant to esterification except with highly reactive acylation partners such as acid chlorides/DMAP; softer electrophiles such as formyl chlorides/DMAP failed to acylate, with the outcome being only a limited qualitative SAR was possible. Since many of the dozen compounds would succumb to esterase hydrolysis at different rates, QSAR was not expected to provide useful results. Fortunately, the Mosher esters of *rac*-**2** could be formed and separated leading to chiral alcohols *R*(−)**-2a** and *S*(+)-**2b**. Of these, the alcohol *R*(−)-**2a** mimics the natural product and is the only alcohol of the pair with antimalarial activity, an important finding in light of previous generalizations regarding the unimportance of chirality for antimalarial activity [16,17]. Clearly, at least in this case, chirality is important to biological activity. Given the probability of protein targets such as *pf*-ATP6 [21,22,23,24,25] and, in the antimalarial effects of peroxides, the issue of chirality in antimalarial drug design is important.

The corresponding 5-ketone, **15**, holds potential for derivation of both C-4 and C-5 although we were unable to methylenate **15** to give **16**. Perhaps this or some other idea can be accomplished in the future to access analogs of artemisinin at the C4/C5 positions via ketone **15**. The antimalarial activity of ketone **15** was substantially improved over the corresponding alcohol **2** and interestingly showed a different ultimate rearrangement pathway under the influence of ferrous ion. The greater antimalarial effect of the ketone **15** relative to the alcohol **2** may be related to the more stable α-keto radical intermediate **23** compared to the more reactive β-hydroxy radical intermediate **21** which undergoes self-immolation rather than reside long enough for intermolecular processes to occur.

## Data Availability

The data presented in this study are available on request from the corresponding author.

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
