# Peer review of "Structure–Activity Relationships of the Antimalarial Agent Artemisinin 10. Synthesis and Antimalarial Activity of Enantiomers of rac-5β-Hydroxy-d-Secoartemisinin and Analogs: Implications Regarding the Mechanism of Action"

_molecules, 2021, doi:10.3390/molecules26144163_

Round 1

Reviewer 1 Report

I would like to congratulate the authors for their work. The methodology and procedures have presented  in detail and are well documented. I wonder if you have performed any docking experiment with target protein with the most active compounds such as the artemisinin-like enantiomer R-(-)-2a.   

Author Response

Reviewer 1 seems to find our manuscript acceptable as it is, perhaps with a spell-check to be sure we didn't miss something.  I have to say that this review was very agreeable.

The reviewer was interested to see if analog 2a had been docked to our protein model of pfATP6.  I am happy to report that these results paralleled the bioassay, i.e. 2a docked more favorably than the unnatural analog.   Furthermore, the manuscript is being formatted for Molecules and should be submitted soon.  Thank you for your interest!

Reviewer 2 Report

Figure 1: I suggest changing the dotted line to a dashed wedge to specify that the bond is
made behind the paper plane. In an analogous way to that used in scheme 1.
Line 80: change “Desmethyl” to “desmethyl”
General: I suggest not using the term "equiv" as, in general, it can cause confusion in many
cases. I prefer the quantities to be expressed, for example, using a molar relationship.
Line 335: Faced with numerous possibilities for performing energy calculations, basing the
discussion on energy differences calculated through the obsolete MM2 method seems to me
simply unacceptable.
Line 348: Once again, it seems to me unacceptable to use MM2 to support the discussion of
the work.
Line 372: “High resolution mass spectra (HRMS)”. Conditions used ?
Line 386: “4.1.2.1. Ethyl 2-(2-methyl-1,3-dioxolan-2-yl) acetate (3)”. Are the experimental
data (NMR, IR etc.) obtained in agreement with the same data reported in the literature?
Why was the reference with such data not cited?
The questions above, presented for compound 3, are valid for all other compounds.

Author Response

for reviewer 2

Fig. 1: revised

Line 80: changed, and all equiv changed to molar equivalents

Line 335, 348: we redid all the calculations using DFT and updated with figures 3 and 4 as well as the text

Line 372: We used language for the HRMS consistent with other molecules papers.

Line 386: References were added for known compounds.

We thank Reviewer 2 for time and effort, and helpful comments

Reviewer 3 Report

Malaria continues to be one of the parasitic diseases.

But now the development of new drugs requires consideration of the structure of many known active compounds. For example virtual screening.

However, the article is useful in terms of synthesising new compounds and expanding the datasets of measured activities against malaria.

Author Response

There did not seem to be any points to respond to, but we thank reviewer 3 for the time and effort spent helping us.

Reviewer 4 Report

The authors of the work “Structure-Activity Relationships of the Antimalarial Agent Artemisinin 10. Synthesis and Antimalarial Activity of Enantiomers of rac-5β-Hydroxy-D-Secoartemisinin and Analogs: Implications Regarding the Mechanism of Action” made and interesting work. Nevertheless, some changes must be done in order to be publish.

  1. First, the aspect of Structure Activity Relationship must be implemented, specially with the use of computational chemistry methods. Since they focused their discussion on the electronic aspects of Artemisinin derivatives, they need to use a more rigorous and precise theoretical approach, like Density Functional Theory or Hartree-Fock.
  2. I suggest that the authors to into consideration the calculation of chemical reactivity descriptors (global and local) in order to study the effect over the trioxane moiety. I suggest the following works as a reference: https://doi.org/10.1016/j.bmc.2018.10.045 and https://doi.org/10.1016/j.molstruc.2020.128456.
  3. Finally, I suggest that if the authors have the possibility, it will be interesting to consider the fragmentation patterns of these compounds, since their mechanism of action is related to the stability and the breaking of the bond between oxygen atoms. I suggest this work as a reference: 1007/s11356-020-10071-0

Author Response

Reviewer 4

  1. We converted all calculations to DFT as outlined in the text.  Many new references were added, but the dataset was too small for a QSAR study. 
  2. Implementing QM descriptors to do even a small QSAR is somewhat outside the scope of this article.  Originally, we had hoped that the alcohol 2 would be reactive, but in fact, it took us years of effort to make a dozen compounds.  We did refer to as many papers as we thought relevant to artemisinin QM descriptors and cited the work of Santos, et al., as an example.
  3. I agree that looking at fragmentation patterns in MS is an excellent idea, but would be a paper on its own.  This reviewer seems to be familiar with Artemisinin.  I suggest that a dataset could be culled from the literature in order to obtain a large set of structurally variable trioxanes, their MS frag patterns analyzed, and correlated e.g. with QM descriptors.

This Reviewer 4 was particularly challenging and helpful.  From the review, we derived a new Fig3, modified the text, reviewed the literature regarding QM desciptors in artemisinin QSAR, and added numerous new references.  We wish to thank this reviewer for many thoughtful comments that improved our manuscript.